# Systematic Review of EEG-Based Imagined Speech Classification Methods

**DOI:** 10.3390/s24248168

**Published:** 2024-12-21

**Authors:** Salwa Alzahrani, Haneen Banjar, Rsha Mirza

**Affiliations:** Department of Computer Science, Faculty of Computing and Information Technology, King Abdulaziz University, Jeddah 21589, Saudi Arabia; hrbanjar@kau.edu.sa (H.B.); rmirza@kau.edu.sa (R.M.)

**Keywords:** brain–computer interfaces, BCI, inner speech, imagined speech, electroencephalogram, EEG

## Abstract

This systematic review examines EEG-based imagined speech classification, emphasizing directional words essential for development in the brain–computer interface (BCI). This study employed a structured methodology to analyze approaches using public datasets, ensuring systematic evaluation and validation of results. This review highlights the feature extraction techniques that are pivotal to classification performance. These include deep learning, adaptive optimization, and frequency-specific decomposition, which enhance accuracy and robustness. Classification methods were explored by comparing traditional machine learning with deep learning and emphasizing the role of brain lateralization in imagined speech for effective recognition and classification. This study discusses the challenges of generalizability and scalability in imagined speech recognition, focusing on subject-independent approaches and multiclass scalability. Performance benchmarking across various datasets and methodologies revealed varied classification accuracies, reflecting the complexity and variability of EEG signals. This review concludes that challenges remain despite progress, particularly in classifying directional words. Future research directions include improved signal processing techniques, advanced neural network architectures, and more personalized, adaptive BCI systems. This review is critical for future efforts to develop practical communication tools for individuals with speech and motor impairments using EEG-based BCIs.

## 1. Introduction

Scientists worldwide have attempted to classify imagined speech from brain signals using various methods. These studies can potentially help impaired speech reach new horizons in human–computer interactions. One of the main tools used in research on imagined speech is electroencephalography (EEG), a non-invasive technique that measures brain activity using scalp electrodes. It has evolved into one of the main instruments in imagined speech studies owing to its simplicity, portability, and safety [1].

Imagined speech is a high-level neural system-based communication method that allows people with severe speech or motor disabilities to link their brains to the environment. The concept of imagined speech has rapidly evolved over the recent decade. The origin of EEG research can be traced back to 1928, when Hans Berger made landmark strides in the field. This soon led to the development of sophisticated computational models for classifying EEG data, which exist to date [2].

Most recent research has defined the general concept of imagined speech well under research conditions, including the entire pipeline from signal acquisition to preprocessing and further analysis. During the preprocessing step, different filters and methods, such as band-pass filtering, notch filtering, Independent Component Analysis (ICA) [3] for artifact removal, and Discrete Wavelet Transform (DWT) [4], are usually implemented. Thus, providing some improvement within EEG-based imagined speech classification, these methods provide a better means of brain–computer interface (BCI) [5].

The evolution of computational methods has significantly shaped the field of imagined speech classification. Initially, standard classification techniques such as Common Spatial Patterns (CSPs) and Support Vector Machines (SVMs) [6] were widely used. Over time, these methods were enhanced with more sophisticated approaches like Sparse Logistic Regression (SLR) [7], Extreme Learning Machines (ELMs) [8], and Linear Discriminant Analysis (LDA) [9], resulting in improved accuracy. Modern advances have seen the integration of deep learning methods, which have revolutionized the field by automating feature extraction and introducing more robust classification architectures. For instance, Convolutional Neural Networks (CNNs) and Recurrent Neural Networks (RNNs) [3] have had a notable impact on multiclass classification tasks. Recent studies have demonstrated the capability of deep learning to detect diverse imagined speech signals, such as CNNs fused with Temporal Convolutional Networks (TCNs) [10], pushing the boundaries of practical applications for individuals with speech and motor disabilities.

While progress has been significant, several gaps in the literature underscore the need for a focused review of this field. A few similar review papers have been published, each providing valuable insights. For example, the paper [1] provides a foundational overview of EEG-based imagined speech decoding and discusses recent research developments. However, it lacks an in-depth exploration of specific challenges, such as subject-independent approaches and multiclass scalability, which are critical for developing robust and scalable brain–computer interface systems. Another paper [5] provides a broad comparison of EEG with other neuroimaging modalities, such as MEG and fMRI, highlighting their respective advantages and limitations in speech decoding. However, it does not delve into detailed evaluations of feature extraction or classification methods specifically designed for imagined speech decoding tasks. Instead, its focus remains on the general applicability of different modalities rather than the technical intricacies of EEG-based imagined speech classification. Lastly, ref. [11] focuses on deep learning applications, particularly the use of methods like CNNs and RNNs in speech imagery decoding. However, it does not sufficiently address challenges related to brain lateralization or the limitations of existing datasets, such as their size, diversity, and quality. These reviews, while valuable, do not provide a comprehensive examination of methodologies tailored to directional word classification, nor do they emphasize recent advancements in the scalability of BCI systems or the practical applications of EEG-based imagined speech decoding.

The purpose of this study is to address these gaps by systematically reviewing the latest advancements in EEG-based imagined speech classification between 2018 and 2023. It focuses on studies utilizing publicly available EEG datasets, particularly those featuring directional words such as “up”, “down”, “lef”, “right”, “forward”, and “backward”. These directional words are pivotal for developing intuitive and efficient BCIs that facilitate navigation and communication for individuals with severe motor disabilities. Unlike previous reviews, this paper uniquely emphasizes the methodological and practical challenges associated with these datasets, providing a focused analysis of state-of-the-art techniques and limitations.

By examining signal acquisition protocols, preprocessing techniques, feature extraction methods, and classification approaches, this review aims to identify the most effective strategies for achieving accurate imagined speech recognition. The scope is intentionally restricted to studies using publicly available datasets to ensure consistency in methodology comparison and result validation, enabling standardized benchmarking across different studies.

A significant focus of this review is addressing challenges in generalization and scalability. Subject-independent approaches, which eliminate the need for individualized training, are critically analyzed alongside multiclass scalability, essential for the broader application of BCIs in real-world settings. Additionally, this review explores the phenomenon of brain lateralization in imagined speech and its implications for EEG signal processing, contributing to a deeper understanding of neural mechanisms and their role in improving classification performance.

This review also emphasizes the importance of diversity in training data and its role in enhancing the generalizability of BCI systems [12]. It highlights limitations in existing datasets, such as insufficient size, diversity, and annotation quality, which constrain the scalability of classification methods [13]. By addressing these limitations, this study advocates for advancements in dataset development and more rigorous methodological standards.

Ultimately, this review establishes a comprehensive framework for EEG-based imagined speech classification by providing a detailed comparison of feature extraction techniques, classification methods, and performance metrics. It aims to steer the field toward the development of adaptive, precise, and universally applicable BCIs. These advancements promise to improve communication and quality of life for individuals with severe motor disabilities through assistive technologies that leverage accurate classification of imagined speech from EEG signals.

## 2. Materials and Methods

### 2.1. Protocol

This systematic literature review follows a structured and transparent approach to synthesize evidence on EEG-based imagined speech classification.

### 2.2. Eligibility Criteria

The inclusion and exclusion criteria were designed to target studies utilizing open-source EEG datasets that were used to classify directional words. This results from the increasing attention given to EEG in creating BCIs and various assistive technology applications. The criteria also ensured a consistent methodological framework and comparability across studies, thereby enhancing the validity and reliability of our systematic review.

#### 2.2.1. Inclusion Criteria

(a)Focus on EEG-based imagined speech classification:EEG has been proven to be an effective and non-invasive way of recording brain signals. It also has more applications in imagined speech classification, indicating its outstanding contribution to BCIs and neuroprosthetics. This review focuses on EEG for the specificity of better technologies that deal with more relevant technologies in the field.(b)Utilization of public datasets:This ensures that the results are reproducible from one researcher to another using the same raw data. Public datasets also enable a broader comparison of different studies, thereby improving the validity of systematic reviews.(c)Investigation of directional word classification:The use of directional words like “up”, “down”, “left”, “right”, “forward”, and “backward” will give an oriented and measurable imagined speech classification. This focus is particularly relevant for applications in assistive technologies where simple commands are crucial.

#### 2.2.2. Exclusion Criteria

(a)Non-English language studies:Limiting this review to English language studies recognizes the reviewers’ language capabilities. In addition, it ensured a uniform interpretation and analysis of the studies.(b)Studies not using EEG for signal collection:As this is a review specifically on EEG-based classification, other methods such as fMRI or PET scans are not included. This ensures that the review remains focused on related literature without dilution.(c)Scope not focused on classifying imagined speech:This guarantees that the review will be perfectly tailored to the particular scope of interest (imagined speech classification), excluding broader EEG studies on different applications.(d)Studies utilizing classifying vowels, phonemes, letters, or syllables:By omitting studies dealing with vowels, phonemes, letters, and syllables, this review restricts its scope to a particular domain—the classification of directional words. This aids in creating a more cohesive and targeted review.

### 2.3. Information Sources and Search Strategy

A systematic literature search was performed using three electronic databases: Web of Science, PubMed, and Scopus. The decision to use these databases was strategic and guided by the recognition of their extensive coverage and reliability in biomedicine and technical sciences. Such coverage allows access to a wide range of peer-reviewed journals and conference papers relevant to the research area. The time frame for the literature search was precisely determined to cover publications from January 2018 to 6 October 2023. Therefore, this period was selected to ensure the coverage of the latest research information and significant work within the previous few years. The search focused on identifying scholarly articles and conference papers on EEG-based imagined speech. Various combinations of keywords and terminologies that are widely used in this area were included in the search. This method was intended to capture the broadest possible spectrum of various studies. The specific search combinations used were as follows:(covert speech EEG) OR (covert speech Electroencephalography)(decoding speech EEG) OR (decoding speech Electroencephalography)(envisioned speech EEG) OR (envisioned speech Electroencephalography)(imagery speech EEG) OR (imagery speech Electroencephalography)(imaginary speech EEG) OR (imaginary speech Electroencephalography)(imagined words EEG) OR (imagined words Electroencephalography)(inner speech EEG) OR (inner speech Electroencephalography)(silent speech EEG) OR (silent speech Electroencephalography)(unspoken words EEG) OR (unspoken words Electroencephalography)

All combinations were carefully selected to align different facets and interpretations of imagined speech and its relationship with EEG, which ensured that the search was comprehensive and included a wide range of related studies.

### 2.4. Study Selection and Data Management

In this systematic review, a systematic selection process and a data management process were employed to guarantee the integrity and quality of the review. This process involves several key steps:(a)Initial screening:The titles and abstracts of articles found in the databases were assessed during the initial screening. The goal was to identify studies that met the predefined eligibility criteria.(b)Application of predefined eligibility criteria:Predetermined eligibility criteria were applied systematically to ensure an unbiased and straightforward study selection process.(c)Full-text assessment for eligibility:The full text of studies was reviewed to determine their eligibility when titles or abstracts indicated potential relevance or uncertainty. This step ensured comprehensive alignment with the inclusion criteria for the review(d)Data management using Mendeley software (Version 2.100.0) & EPPI-Reviewer software (Version 6.15.1.0):References and data from the selected studies were organized and managed using specialized software tools, including Mendeley and EPPI-Reviewer.Mendeley [14]: A reference manager facilitates the storage, organizing, noting, sharing, and citation of research data and references.EPPI-Reviewer [15]: EPPI-Reviewer manages and analyzes literature review data. While its features apply to any literature review, the software was designed to support various forms of systematic reviews (e.g., framework synthesis and thematic synthesis). The software facilitates qualitative and quantitative analyses, including thematic synthesis and reference management. Additionally, it includes “text mining” technology, which can improve the effectiveness of systematic reviewing.This methodology emphasizes commitment to a thorough, elaborate strategy, which is the core factor in the systematic review’s robustness.

### 2.5. Data Collection Process

The data collection process was designed to acquire comprehensive information from the selected studies. Therefore, captured data will facilitate analysis and comparison across different studies. The information extracted included the following:(a)Author(s’) information: The authors of each study were identified to recognize their contributions and understand the diversity of the research community.(b)Publication year: The year of publication was recorded to evaluate the timeline and progression of research in the field.(c)Research objectives: The specific objectives of each study were described within the scope of EEG-based imagined speech classification, highlighting the diverse approaches used for classifying directional words.(d)Reported limitations: The limitations reported by the authors were documented, including methodological challenges, data handling difficulties, issues with generalizing classification models, and other obstacles encountered in this area of research.(e)Suggested future directions: Future research avenues proposed by each study were captured to highlight potential advancements in algorithm development and EEG signal processing techniques. This information is critical for identifying opportunities in the field and guiding subsequent research efforts.(f)Utilized dataset details: Details of the datasets used, including the subjects, devices, and protocols, were collected to enhance the understanding of the relevance and applicability of the research findings.(g)Signal processing techniques: The techniques used for raw data processing were documented to provide insights into the initial steps of data handling.(h)Data downsampling: Information on whether the data were downsampled to reduce its size was documented.(i)Feature extraction methods: The methods used to identify and extract relevant features from the data were documented, as they are essential for effective data classification.(j)Classification methods: The methods used for classifying the data were detailed to facilitate the comparison of results across studies.(k)Optimization strategies: The techniques implemented to optimize classification performance were documented.(l)Reported accuracy: The accuracy reported in each study was recorded to measure the effectiveness of the methodologies employed.(m)Evaluation metrics: Additional metrics used to assess the performance of the classification were documented in detail.(n)Classification type: The type of classification, whether binary or multiclass, was recorded to evaluate its impact on the complexity and effectiveness of the applied methods.(o)Subject dependency: The approach was identified as either subject-dependent (tailored to specific individuals) or subject-independent (generalizable across individuals), as this distinction influenced the scope and applicability of the findings.

By capturing this extensive range of information, the data collection process aimed to develop a comprehensive database that facilitated in-depth analysis and comparison across studies, making a significant contribution to the body of knowledge in the field.

## 3. Results

### 3.1. Study Selection

The study selection process for this research commenced with an initial database search that yielded 1062 records. Specifically, 557 records were retrieved from the Scopus database, 302 from the Web of Science database, and 203 from the PubMed database. After carefully removing the duplicates, 655 unique records remained. Critical screening of titles and abstracts led to the exclusion of 502 records that did not precisely align with the inclusion criteria because of their lack of relevance to EEG-based imagined speech classification. This narrowed the focus to the 153 full-text articles assessed for eligibility. Of the 153 research studies that underwent full research assessment, 141 were eliminated from the study for failing to either use a public dataset in their research or use prompts other than directional words. Ultimately, 12 studies met the inclusion criteria and were considered for the research synthesis. Figure 1 effectively demonstrates the process flow, outlining the steps from the initial pool of records to the final selection of relevant studies.

### 3.2. Synthesis of Results

This part of the systematic review critically synthesizes the findings of selected studies that are focused on EEG-based imagined speech classification, structured around several key aspects of this line of research. Each section was dedicated to a particular area of study. A summary of the reviewed studies, including datasets, preprocessing methods, feature extraction techniques, classification approaches, and accuracy results, is provided in Table 1. This summary allows for a structured comparison of methodologies and performance metrics across studies, facilitating insights into common approaches in EEG-based imagined speech classification.

#### 3.2.1. Effect of Feature Extraction on Classification Performance

Feature extraction is critical for classifying EEG-based imagined speech. The selection and extraction of relevant features from EEG data significantly impact the accuracy, robustness, and discriminative power of classification. The reviewed papers discuss different feature extraction methods and their impact on classification performance.

In a broader view, the research proposed in [16] adopted a unique method of moving forward with the classification of EEG-based imagined speech. It uses multivariate fast and adaptive empirical mode decomposition (MFAEMD) for feature extraction, aiming at specific features, such as slope domain entropy (SDE) and L1-norm features. These extracted features were then classified using various machine learning methods. One crucial methodological advancement is the application of correlation-based feature selection (CFS), which determines the most essential features and decreases the dimensionality of the feature vector. This strategy effectively removes the curse of dimensionality and increases classification accuracy. Nevertheless, it is worth noting that the best accuracy recorded using dictionary learning classification is relatively modest at 21.53% compared to other studies in the field. However, this study is still helpful because it shows the vital role of optimized feature extraction and selection in improving classification performance in BCI systems.

The authors in [17] focused on applying a Siamese Neural Network (SNN) for feature extraction from raw EEG signals. The SNN model is trained to optimize the distance in the embedding space; therefore, it learns discriminative features for the classification of imagined speech. These features were derived from the raw EEG data directly by the SNN model, without any manual feature engineering, which made the system automatic and data-driven when extracting relevant features. The results demonstrate that the SNN model achieves a higher classification accuracy than conventional methods, proving its effectiveness in feature extraction. This indicates that deep learning models can capture the patterns required for improvement during the feature extraction process.

Building on the previous study, the same authors further advanced their research in [18]. This study introduces deep metric learning, which entails extracting features using instantaneous frequency and spectral entropy. Combined with the ability of the SNN to reduce high-dimensional EEG features, such features facilitate a more efficient learning of distances between reduced embeddings. When used with the K-Nearest Neighbors (KNN) classifier, this technique performed better than the method used in the study [17] by achieving higher classification accuracy. This development demonstrates the critical role of sophisticated feature extraction in improving the efficiency and precision of BCI systems.

The authors in [19] suggested the use of smoothed pseudo-Wigner–Ville distribution (SPWVD) and CNN for feature extraction. In contrast to conventional methods such as the Continuous Wavelet Transform (CWT) and Short-Time Fourier Transform (STFT), the SPWVD method provides a more precise discrimination of the time–frequency–energy distribution within EEG signals. Post-processing of SPWVD-based features using the CNN model demonstrated its capabilities in feature extraction and classification in an automated fashion. The results showed that the proposed feature extraction method performed better than the pre-trained CNN models, and, evidently, SPWVD-CNN is an appropriate method for capturing pertinent information to enhance classification accuracy.

The paper [20] explored the concept of feature extraction, which would further allow even better classification of EEG signals for imagined speech. Central to this work is the use of the optimized rational dilation wavelet transform (RADWT), enhanced by particle swarm optimization (PSO), to select the best tuning parameters. This approach allows the adaptive decomposition of EEG signals into several sub-bands, from which several statistical features are computed. The features extracted from EEG signals were passed on to multiple machine learning algorithms for classification. This research proves that feature selection quality contributes to classification accuracy. The results of this study demonstrated that this optimized strategy surpasses the others with respect to the accuracy of classification results, which emphasizes the necessity of effective feature extraction in EEG signal classification and its application in BCI research.

The research work [21] aimed to develop EEG signal classification by decomposing the signal into separate bands of five primary rhythms (delta, theta, alpha, beta, and gamma). The decomposition method applied was DWT, which differs from the traditional feature extraction approach based on spatial location. This approach, using higher-order Daubechies wavelets and further amplified by a bandwise CSP, was extended for multiclass problems to improve signal discrimination. Multiclass classification using SVM with a Radial Basis Function (RBF) kernel was employed. The results, which were validated using five-fold cross-validation, clearly indicate the efficacy of rhythm-specific processing in EEG-based imagined speech classification.

The study by [22] investigated the innovative application of tensor decomposition, known as Parallel Factor Analysis (PARAFAC), to improve EEG classification performance by analyzing imagined speech signals. This work contributes significantly to the analysis of multivariate data techniques that enable a detailed analysis of the complex relationships among different dimensions of EEG signals, such as space, time, and frequency. The use of PARAFAC allows the authors to address a critical gap in previous studies, in which the interconnection between these brain signal dimensions is sometimes neglected. Thus, it provides a more comprehensive perspective on EEG data. This study shows that the application of extracted PARAFAC components enhances the discriminability of imagined words, characterized by greater accuracy and a lower standard deviation across subjects. Thus, it demonstrates not only the efficiency of the method but also its robustness across different individuals. The results of this study highlight the power of advanced feature extraction approaches, such as tensor decompositions, in a more representative and discriminative manner for complex brain signals. However, this work requires optimization of computational power to enhance the performance of a practical imagined speech system.

The reviewed studies highlight the effectiveness of various feature extraction techniques in EEG-based imagined speech classification. Traditional methods like tensor decomposition (PARAFAC) in [22] and DWT in [21] focus on decomposing EEG signals into interpretable components, achieving competitive accuracy, such as 59.7% in [22]. These methods excel in capturing multi-dimensional relationships within the data but often require significant preprocessing effort. In contrast, deep learning-based approaches like SNN [17] and SPWVD-CNN [19] prioritize automated feature extraction, showing improved accuracy at 45% and 60.42%, respectively. These methods highlight the power of deep architectures in identifying patterns without manual feature engineering but demand high computational resources and large datasets. Adaptive optimization techniques, such as RADWT-PSO [20], stand out by balancing efficiency and performance, achieving the highest accuracy at 72.36%. These findings emphasize the importance of tailoring feature extraction methods to the specific characteristics of EEG data and the intended application.

#### 3.2.2. Classification Methods to Enhance Classification Performance

Classification methods are necessary to improve classification performance in many domains. This section describes a review of studies that focused on using classification methods to develop a robust classification of EEG-based imagined speech signals. The goal was to identify the potential of machine learning and deep learning algorithms for classifying imagined speech.

The work in [23] investigated the performance of machine learning algorithms, namely Random Forest (RF), SVM, and KNN, in the classification of EEG signals for imagined speech. The results showed that the SVM outperformed all other methods, with a testing accuracy of 33.9% for a multiclass classification task involving four classes. This success is attributed to its ability to handle the nonlinear nature of EEG data by using a quadratic kernel function. Nevertheless, the overall modest performance only highlights the challenges in EEG signal classification, such as EEG signal complexity and the characteristic nature of imagined speech.

Moving on to [24], which provided a detailed examination of EEG-based imagined speech classification methods. It compared traditional machine learning techniques, such as SVMs, LDA, and RF, with deep learning approaches, especially CNNs. The study focused on the superiority of CNNs to EEG classification, primarily linked to their capacity to simultaneously achieve feature extraction and classification. This research adopted nested cross-validation (nCV) for hyperparameter (HP) optimization and evaluated intra-subject and inter-subject optimization methods. This approach involved testing various HPs, such as activation functions, learning rates, and loss functions for CNNs, as well as the different kernels and regularization parameters for SVMs. As demonstrated by ANOVA tests and statistical analysis, this illustrated the significant effects of HP optimization and, hence, the critical role played by HPs in boosting the performance of the ML and DL classifiers when used in the realm of EEG-based imagined speech.

Shifting our focus to [25], which relies on applying CNNs, precisely the EEGNet architecture, in classifying EEG-based imagined speech signals, as one of its essential deployment areas in BCI research. The study found that deep learning models are competent at drawing pertinent features from raw EEG signals so that imagined speech recognition is possible. Although it has an average modest accuracy of 29.7% in this classification, the research identified the approach as feasible, which brings the following drawbacks: small data size and influence from other brain activities. This paper presented its methodology and results compared to existing studies, contributing to an evolving understanding of the intersection of neuroscience with classification methods.

Furthermore, ref. [26] focused on machine learning and deep learning techniques with regard to the classification of imagined speech. It compares machine learning with deep learning methods on two publicly imagined speech datasets. Several classification methods were studied in this research, from the simplest, such as SVMs, to ensemble methods, which included the powerful eXtreme Gradient Boosting (XGBoost) and the most complicated Long Short-Term Memory (LSTM) and Bidirectional LSTM (BiLSTM) for neural networks. The BiLSTM models displayed better performance owing to their ability to capture sequential dependencies. The effective model consisted of a BiLSTM layer, two dense layers, and a dense output layer incorporating dropout layers to prevent overfitting. Despite the success of the BiLSTM models, the overall classification results suggest that imagined speech classification is still evolving and requires further improvements, including fine-tuning parameters at an individual level and generating more comprehensive datasets with less noise.

The reviewed studies demonstrate a clear contrast between traditional machine learning approaches and deep learning models in EEG-based imagined speech classification. Traditional algorithms such as SVMs [23] and RF offer simplicity and interpretability, achieving accuracies of up to 33.9%. However, these methods struggle with capturing the nonlinear complexity of EEG signals. Deep learning methods like BiLSTM [26] offer a slight improvement with an accuracy of 36.1%, showing their potential to model sequential dependencies in EEG data but also highlighting their limitations when tested on complex datasets. The difference between these two approaches is not substantial, suggesting that more advanced optimization techniques may be required to unlock the full potential of deep learning models in this domain.

Conversely, CNN-based methods [24] demonstrate stronger performance, especially when combined with robust hyperparameter tuning, achieving accuracies of 60.42% in studies such as [19]. However, this improvement often comes at the cost of increased computational complexity. Ensemble methods like Bagging coupled with RADWT [20] surpassed other techniques with the highest accuracy of 72.36%, showcasing the effectiveness of combining feature optimization with ensemble classification strategies.

These findings emphasize that while traditional methods like SVMs remain competitive in certain scenarios due to their simplicity, the marginal improvements from BiLSTM models suggest a need for further optimization. Meanwhile, CNNs and ensemble techniques provide a stronger foundation for addressing the nonlinear and high-dimensional nature of EEG signals, particularly when supported by advanced feature extraction and tuning.

In summary, the reviewed papers presented significant potential for classification methods, including traditional machine learning and deep learning methods, to improve the classification performance of imagined speech. Although some specific classification methods, such as CNNs, have performed with higher accuracy, more enhancements are required to reach the field’s maximum potential.

#### 3.2.3. Investigation of Brain Lateralization in Imagined Speech

Lateralization of the brain is an underlying concept in human cognition and is very important when studying imagined speech. Most precisely, both brain hemispheres divide and lateralize cognitive functions related to language comprehension and production. The left hemisphere is dominant in language processing and controls speech comprehension and production. The Wernicke–Geschwind model further supports the dominance of the left hemisphere, whereby the model causes lateralization of language functions. However, this model points out that the significant areas in the left hemisphere (Wernicke and Broca areas) are responsible for the perception and production of speech, respectively [27].

Imagined speech is an internally verbalized and self-produced speech that is not overtly expressed. This phenomenon involves auditory imagery of an “internal voice” and activates brain areas relevant to comprehension and speech production. Therefore, investigating brain lateralization in imagined speech aims to uncover whether the same left hemisphere regions involved in pronounced speech are also active during the generation of imagined speech.

The study [26] employed diverse computational models such as SVM, XGBoost, and state-of-the-art deep learning architectures ranging from LSTM to BiLSTM. The incorporation of BiLSTM further improves the accuracy level in classifying the imagined words, signifying that the neural network architectures used have a very high potential to increase BCI performance. This study focuses on brain lateralization, which indicates that language-processing activities are localized mainly in the left hemisphere. The research findings indicate no difference between the model’s performance using the left hemisphere is signals and that of the whole brain. Therefore, the left hemisphere alone holds sufficient information to precisely classify imagined speech. This finding supports the concept of brain lateralization in imagined speech. This study is innovative in that it enables a comparison of the classification results of EEG signals from the left and right hemispheres of the brain. An essential investigation component is to compare EEG signal classification results between the left and right hemispheres of the brain.

A related work [27] analyzed brain lateralization related to imagined speech using EEG signal, as per the Wernicke–Geschwind model. This paper introduces the use of more diverse feature extraction techniques, such as DWT, Empirical Mode Decomposition (EMD), and Variational Mode Decomposition (VMD), because these methods play an important role in decoding the intricate nature of EEG signals. Central to this research is the use of SVM classifiers with both linear and nonlinear kernels, gauged in performance by a rigorous five-fold cross-validation method. A critical aspect of this study is the comparison of the results from the classification of EEG signals of the left and right hemispheres of the brain. The results clearly demonstrate more activity in the left dominant hemisphere during covert speech, aligning with the typical speech processes and supporting the predictions of the Wernicke–Geschwind model.

The reviewed studies demonstrate a consistent pattern of left-hemisphere dominance in language processing during imagined speech classification. Both [26,27] affirm the viability of using left-hemisphere signals alone for precise classification, streamlining the computational requirements for BCI systems. While [27] focuses on traditional feature extraction methods like DWT and EMD combined with SVM classifiers, achieving modest yet interpretable results, ref. [26] leverages BiLSTM for sequential modeling, providing slightly enhanced accuracy.

Interestingly, the findings in [26] indicate that using the whole brain does not significantly outperform using only left-hemisphere signals, which supports the concept of lateralization while offering practical implications for EEG channel reduction in real-world applications. Both studies highlight the importance of leveraging neural activity localization for efficient classification, with [27] showcasing the role of feature engineering and [26] emphasizing deep learning models for capturing sequential dependencies.

In summary, lateralizing imagined speech in the brain using EEG signals provides insight into the neural processes that represent imagined speech. Left-hemisphere dominance proves the concept of brain lateralization and its importance in designing and developing efficient imagined speech classification systems. Therefore, further research on this topic is urgently required to fully comprehend and exploit it in practical speech rehabilitation applications.

#### 3.2.4. Generalization and Scalability in EEG-Based Imagined Speech

(a)Subject-independent approach:This approach aims to develop general models that can cover different subjects without individual calibration or training, which are very important for practical implementation and wide use in EEG-based imagined speech systems. The models can be easily applied to new users without extensive training sessions and customization, making the systems usable and scalable.In [16], a subject-independent leave-one-out cross-validation (LOO CV) approach was used for the performance evaluation of classifying imagined words. This is crucial, as it is relevant to understand how well generalizing a BCI system across individuals without individual-specific training is possible. The model in the subject-independent LOO CV was learned from all the subjects, but one was held out as a test set. This was repeated so that each subject’s data were taken as the test set exactly once. In this regard, the testing strategy helps judge the model’s generalization capability across new and unfamiliar subjects. The results showed that although classifiers could handle subject variability up to a certain level, the performance metrics simultaneously showed room for improvement in accuracy and reliability across unseen subjects. Thus, while the subject-independent LOO CV approach widens the applicability of BCI systems, high accuracy remains a challenge and an open area for future research.In contrast, the work in [19] provides a rigorous testing framework in which models are trained and tested across different groups of subjects to ensure effective generalization. The importance given to subject-independent generalization indicates that the generalization of BCI systems for recognizing imagined speech can be effective for different subjects without adaptation to the individual subject. Such a development is necessary to generalize and practically deploy BCI technologies in real-life applications. The results obtained from this dataset show that the model is generalizable across subjects with high accuracy. This eliminates the need for detailed recalibration with new users.The paper [20] also presented a subject-independent approach that uses all collected data for training and testing models without dividing the data into subgroups by individual subjects. Therefore, the results discussed in this paper show that a subject-independent approach attains considerably high accuracy rates, thus demonstrating the ability of a BCI system to do so. The performance also outperformed other studies, revealing robust generalization ability and capturing variations across different individuals as needed in a scalable BCI.The reviewed studies underscore the critical trade-offs in subject-independent approaches. While the LOO CV approach in [16] provided foundational insights into generalization, its relatively low accuracy (21.53%) reflects the difficulty in adapting models to diverse EEG patterns. In contrast, the approaches by [19,20] achieved significantly higher accuracies of 60.42% and 72.36%, respectively, demonstrating the benefits of advanced optimization and robust evaluation frameworks. These findings suggest that subject-independent approaches are increasingly viable, with optimization techniques playing a pivotal role in closing the performance gap with subject-dependent models.(b)Multiclass scalability:One of the most significant hurdles in advancing BCI systems for imagined speech is the ability to scale to new classes. Typically, adding a new class to a pre-existing classifier requires collecting a substantial amount of new data for all classes to retrain the system. This process is time-consuming and impractical for dynamic and real-world applications.The study by [18] innovatively employed deep metric learning to facilitate this scalability, allowing a system to be trained on a set number of classes to accommodate additional classes without requiring comprehensive retraining. This approach utilizes incremental learning, where a classifier that is already proficient in distinguishing between established classes is fine-tuned to recognize new classes by adjusting only its fully connected layers. Crucially, this method preserves the classifier’s performance on the original classes, effectively broadening the system’s capabilities without compromising the existing knowledge base. The success of this approach in maintaining accuracy while adding new classes suggests a path forward in which BCI systems can evolve and grow significantly.

#### 3.2.5. Performance Benchmarking

The selected studies for the systematic review considered two public datasets [28,29]. The benchmark offers insights into the variability of classification accuracies and the influence of dataset characteristics and model sophistication on them.

The open-access database [28] consists of EEG recordings from 15 subjects, each imagining the pronunciation of the Spanish words “up”, “down”, “left”, “right”, “forward”, and “backward”. Each subject repeated each word 50 times in a different order to obtain a rich dataset for analysis. The participants sat in a sound-attenuated chamber facing an LCD monitor showing the visual stimuli, accompanied by an auditory stimulus through the headphones. This setup attempted to impose standardized recording conditions across all participants. EEG data were obtained using a six-channel acquisition system with a sampling frequency of 1024 Hz. To eliminate artifacts, the preprocessing step involves an FIR pass-band filter of EEG signal in the 2 Hz to 40 Hz frequency range. Naturally, the most crucial aspect of quality preservation in data are the removal of artifacts, particularly those of muscle activity, electrode pops, and environmental noise. Data that included such artifacts were omitted to maintain the quality of EEG patterns related to imagined speech.

On the other hand, the dataset [29] encompasses EEG recordings from ten participants. These subjects were unknown to the domain of BCI and conducted various mental tasks under several conditions, namely inner speech, pronounced speech, and visualized conditions. In this case, each participant recorded a single-day session, which comprised an average between 475 and 570 trials. Specifically, data were acquired via the BioSemi ActiveTwo system with 128 active EEG channels and 8 active external EOG/EMG channels at a sampling frequency of 1024 Hz. A standard EEG cap was placed on each participant for optimal electrode placement and conductance with the help of conductive gel. Artifact rejection was performed as a fundamental part of the processing pipeline, in which the EOG/EMG channel signals were used to identify and exclude from the analysis those trials that were substantially contaminated by related muscular activity, particularly from facial muscles. This maintains the integrity of the data so that only the neural activity related to the speech task is captured, avoiding any contamination from physical movements.

Moving on to studies that employed the dataset [28], there has been a significant spread in the reported accuracies. The application of dictionary learning models [16] yielded a binary classification accuracy hovering around the 60% mark, which is relatively low for binary tasks and indicates limited effectiveness in separating the two classes. For the more complex six-class classification scenario, the performance was even lower, with an accuracy of 21.53%. However, more advanced approaches, such as those reported by [20] in ARADWT-driven systems, achieved a much higher accuracy of over 72%, indicating that sophisticated algorithms might outperform simple models by a considerable margin. In turn, the CNN-based methods in [17,18,24] demonstrated a vast span in performance, from relatively modest accuracies of approximately 24% to more substantial ones of approximately 45%, indicating the importance of network architecture and learning protocols in generating meaningful patterns from EEG data.

The results for the dataset [29] were constrained, with accuracies typically falling in the mid-30s range. The SVM model presented in [23] achieved a classification accuracy of 33.9%, which aligns closely with the BiLSTM model proposed in [26], demonstrating a slightly higher accuracy of 36.10%. These findings highlight the potential of both classical machine learning techniques and recurrent neural network architectures for interpreting complex datasets. However, these numbers remain modest compared to the higher performances achieved with the dataset [28], suggesting intrinsic differences in dataset complexity or task formulation. Figure 2 visualizes the comparison of classification methods and feature extraction techniques across datasets, with the highest accuracies achieved using ARADWT for feature extraction and bagging for classification.

Furthermore, when comparing the subject-dependent approach with the subject-independent approach using the same dataset, the box plot in Figure 3 illustrates notable differences in accuracy distribution. The bars represent the range of accuracies achieved across studies, with the upper and lower whiskers indicating the maximum and minimum values, respectively, while the shaded box highlights the interquartile range (IQR). The dots within the box and outside the whiskers represent individual study results, showing the variation in accuracy for each approach. The subject-independent approach exhibits a wider spread, reflecting greater variability, and includes both the highest and lowest accuracy values observed, indicating its potential for higher performance but also its inconsistency across studies. In contrast, the subject-dependent approach demonstrates a narrower accuracy range, suggesting more stable performance within individual subjects, but it lacks the scalability and adaptability of subject-independent approaches, which are essential for broader practical applications.

In conclusion, the summary provided in Table 1 consolidates the key findings and methodologies from the studies included in this review. By detailing datasets, preprocessing methods, feature extraction techniques, classification approaches, and evaluation metrics, the table offers a comprehensive overview of the current state of EEG-based imagined speech classification research. This synthesis highlights the variability in performance across studies, reflecting differences in dataset characteristics, methodological choices, and classification strategies. The table serves as a valuable reference for identifying trends, challenges, and opportunities for advancing the field, ultimately guiding future research toward more robust and scalable brain–computer interface systems.

**Table 1 sensors-24-08168-t001:** Summary of EEG-based imagined speech classification studies.

Paper	Dataset	Downsampling	Preprocessing	Feature Extraction	Classification Methods	Classification Type (Number of Classes)	Optimization	Evaluation	Accuracy	Subject Dependency
Berg et al. (2021) [25]	Nieto et al. (2022) [29]	Downsampled to 254 Hz	Bandpass filter (0.5–100 Hz), Notch filter (50 Hz), ICA	No feature extraction	CNN EEGNet	Multiclass classification (4 classes)	Not mentioned	Accuracy, F1-score, Recall, Precision, K-fold cross-validation	29.67%	Subject-dependent
Biswas et al. (2018) [27]	Coretto et al. (2017) [28]	Downsampled to 128 Hz	No preprocessing	RWE, VMD	SVM-rbf	Multiclass classification (6 classes)	Tuning parameter Y for SVM-rbf	Accuracy K-fold cross-validation	33.63%	Subject-dependent
Biswas et al. (2022) [21]	Coretto et al. (2017) [28]	Not mentioned	No preprocessing	DWT, CSP	SVM-rbf	Multiclass classification (6 classes)	Not mentioned	Accuracy, K-fold cross-validation	40.59%	Subject-dependent
Cooney et al. (2020) [24]	Coretto et al. (2017) [28]	Downsampled to 128 Hz	Bandpass filter (2–40 Hz), ICA with Hessian approximation preconditioning	No feature extraction	CNN EEGNet	Multiclass classification (6 classes)	nCV, Hyperparameter Selection (activation function, learning rate, epochs number, loss function)	Accuracy, Precision, ANOVA, nCV, Tukey Honest Significant Difference (HSD) Test	24.97%	Subject-dependent
Dash et al. (2022) [16]	Coretto et al. (2017) [28]	Not mentioned	Not mentioned	MFAEMD, SDE, L1-Norm, Correlation-based feature selection (CFS)	Dictionary Learning	Multiclass classification (6 classes)	Grid search optimization	Accuracy, F-score, LOO CV	21.53%	Subject-independent
García-Salinas et al. (2018) [22]	Coretto et al. (2017) [28]	Downsampled to 128 Hz	Not mentioned	CWT, PARAFAC, K-means clustering, Histogram generation	SVM-linear	Multiclass classification (6 classes)	Not mentioned	Accuracy, K-fold cross-validation, ANOVA	59.7% (For the first three subjects only)	Subject-dependent
Gasparini et al. (2022) [26]	Coretto et al. (2017) [28]	Not mentioned	Bandpass filter (2–40 Hz)	No feature extraction	CNN BiLSTM	Multiclass classification (6 classes)	Stochastic gradient descent (SGD), nCV	Accuracy	25.1%	Subject-dependent
Gasparini et al. (2022) [26]	Nieto et al. (2022) [29]	Downsampled to 254 Hz	Bandpass filter (0.5–100 Hz), Notch filter (50 Hz)	No feature extraction	CNN BiLSTM	Multiclass classification (4 classes)	Stochastic gradient descent (SGD), nCV	Accuracy, F1-score, Recall, Precision	36.10%	Subject-dependent
Kamble et al. (2022) [19]	Coretto et al. (2017) [28]	Not mentioned	Not mentioned	SPWVD	CNN	Multiclass classification (6 classes)	Hyperparameter optimization (KerasTuner library)	Accuracy	60.42%	Subject-independent
Kamble et al. (2023) [20]	Coretto et al. (2017) [28]	Not mentioned	Not mentioned	ARADWT, Statistical features, ANOVA	Bagging	Multiclass classification (6 classes)	nCV, PSO	Accuracy	72.36%	Subject-independent
Lee et al. (2020) [17]	Coretto et al. (2017) [28]	Downsampled to 128 Hz	No preprocessing	SNN	KNN	Multiclass classification (6 classes)	ADAM optimizer	Accuracy, F1-score, Recall, Precision, K-fold cross-validation, ANOVA, Paired *t*-test	31.4%	Subject-dependent
Lee et al. (2021) [18]	Coretto et al. (2017) [28]	Not mentioned	No preprocessing	SNN, Spectral entropy, Instantaneous frequency	KNN	Multiclass classification (6 classes)	ADAM optimizer	Accuracy, K-fold cross-validation, ANOVA analysis, *t*-test	45%	Subject-dependent
Merola et al. (2023) [23]	Nieto et al. (2022) [29]	Not mentioned	Bandpass filter (0.5–100 Hz), Notch filter (50 Hz)	Statistical features, ANOVA	SVM-quadratic	Multiclass classification (4 classes)	Grid search optimization	Accuracy	33.9%	Subject-dependent

#### 3.2.6. Limitations

The 12 research papers covered in this literature review on imagined speech classification in EEG signals contribute significantly to the methodologies, techniques, and models of the respective imagined speech classifications. Deep learning, machine learning, and other signal processing methodologies hold promise and may help classify imagined speech from EEG.

Some limitations and open issues are identified in this review. For example, studies mention that more diversified and extensive data are needed to validate and improve their models [16,19,23,24]. They emphasized the importance of overcoming the risk of overfitting by expanding the dataset and conducting rigorous validation. The papers also suggested exploring additional feature extraction methods to capture more meaningful information from EEG signals [22,23,25]. This includes investigating different decompositions, such as EMD and wavelet decompositions, and utilizing rhythm-specific spatial features.

Furthermore, these papers highlight the importance of considering practical implementations in real-time BCI systems. This involves developing faster analysis methods, optimizing hyperparameters, and reducing dimensionality to ensure real-time performance [22]. These studies also stressed the need for more controlled recording protocols and personalized feature selection to improve the accuracy and robustness of the models [23].

In conclusion, this literature review highlights the advances made in imagined speech classification using EEG signals. These limitations offer valuable insights for further research and development. By addressing these limitations and exploring future directions, researchers can significantly improve the accuracy, performance, and practicality of imagined speech classification using EEG signals.

## 4. Discussion

This systematic review has outlined the current state of the art in EEG-imagined speech classification approaches, putting special emphasis on the analysis of directional words within the broader context of brain–computer interface applications. Large advances within the field are indicated, pinpointing challenges that remain constant in the performance of models, generalization of subjects, and possible real-world uses. By linking these insights into broader implications for the field, this discussion underlines certain areas of interest for future research.

### 4.1. Feature Extraction and Classification Performance

This review highlights the crucial role of feature extraction techniques in achieving high classification accuracy. Advanced feature extraction methods, including SPWVD and MFAEMD, combined with deep learning models, like CNNs and SNNs, hold great promise toward extracting meaningful patterns from complex EEG data. In specifics, approaches like ARADWT combined with optimization strategies such as particle swarm optimization have presented accuracy levels as high as 72.36%. However, the variability in performance across different feature extraction methods calls for further refinement. Due to the nature of the EEG signals, which is non-stationary, improvement in consistency, especially for multiclass tasks, is highly required; thus, future research efforts should be directed at designing more adaptive neural architectures able to capture long-term dependencies within the data.

### 4.2. Generalization Challenges and Subject Independence

The reviewed studies highlight the variability in performance between subject-dependent and subject-independent approaches to classification. Contrary to conventional assumptions, a subject-independent approach can achieve higher accuracy in certain cases, particularly when using datasets like Coretto et al. (2017) [28]. However, this approach also exhibits greater variability, reflecting challenges in achieving consistent performance. Subject-dependent approaches, while typically less variable, may not always outperform subject-independent approaches and are limited by their reliance on individualized calibration, which restricts their scalability for practical applications.

### 4.3. Insights from Brain Lateralization

The findings of this review confirm the involvement of brain lateralization during imagined speech, which is in line with previously developed models, like the Wernicke–Geschwind framework. Secondly, regions from the left hemisphere are consistently found across different studies. This forms a basis for the optimization of BCI models by narrowing the focus toward these important regions. Most importantly, this insight is especially highly relevant to the improvement of spatial resolution for EEG classifiers.

### 4.4. Standardization of Data Quality and Preprocessing

One of the recurring themes in the review is that there is a need for high-quality, standardized EEG datasets that can support both the training and validation of a model. Differences in preprocessing protocols, such as how to reject artifacts or filter frequency bands, seriously affect the comparability and reproducibility of results across studies. Consistency on this matter is highly essential to develop robust BCI models that can perform consistently across diverse datasets and subject groups. In this regard, future studies should focus on the establishment of standardized preprocessing pipelines using adaptive filtering techniques to address intrinsic noise in the data for higher reliability in imagined speech classification models.

### 4.5. Directions for Future Research

Certain directions are identified, on which future research can further build upon the review. The review highlights several areas where further research could advance the field. Key among these is the exploration of hybrid learning models that integrate the strengths of different learning paradigms—such as transfer learning for adapting pre-trained models to new datasets—and hybrid neural architectures that combine convolutional and recurrent layers to capture complex temporal patterns. Moreover, the development of models that incorporate incremental learning could allow the BCI systems to easily expand for new classes or commands without extensive retraining, hence increasing their flexibility in real-world applications.

## 5. Conclusions

This review addresses progress and challenges in EEG-based imagined speech classification, mainly on the classification of directional words. While recent methodologies incorporating advanced feature extraction and machine learning models show promise, challenges such as variability in EEG signals, model generalizability, and real-time adaptability persist.

Future work in this field should thus aim at developing adaptive BCI systems that cater for user variability, achieve high classification accuracy, and allow for scalability to enable practical applications. It will be by addressing these complexities that we will have the opportunity to move one step closer toward realizing BCIs that can actively offer efficient means of communication to speech- and motor-impaired patients to assist them and transform their interactions and autonomy through technology.

## Figures and Tables

**Figure 1 sensors-24-08168-f001:**
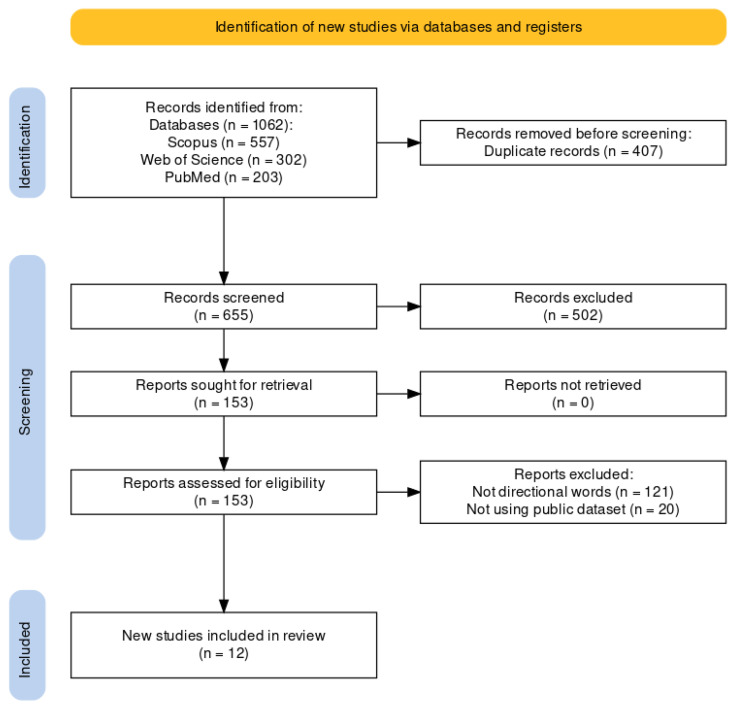
Study selection process.

**Figure 2 sensors-24-08168-f002:**
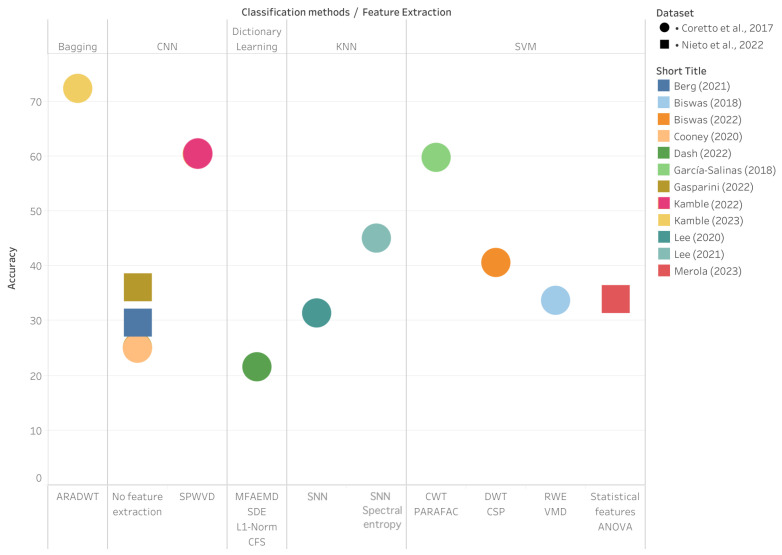
Accuracy plot across all papers using the datasets Nieto et al. (2022) [29] and Coretto et al. (2017) [28]. The results are derived from the following studies: Berg et al. (2021) [25], Biswas et al. (2018) [27], Biswas et al. (2022) [21], Cooney et al. (2020) [24], Dash et al. (2022) [16], García-Salinas et al. (2018) [22], Gasparini et al. (2022) [26], Kamble et al. (2022) [19], Kamble et al. (2023) [20], Lee et al. (2020) [17], Lee et al. (2021) [18] and Merola et al. (2023) [23].

**Figure 3 sensors-24-08168-f003:**
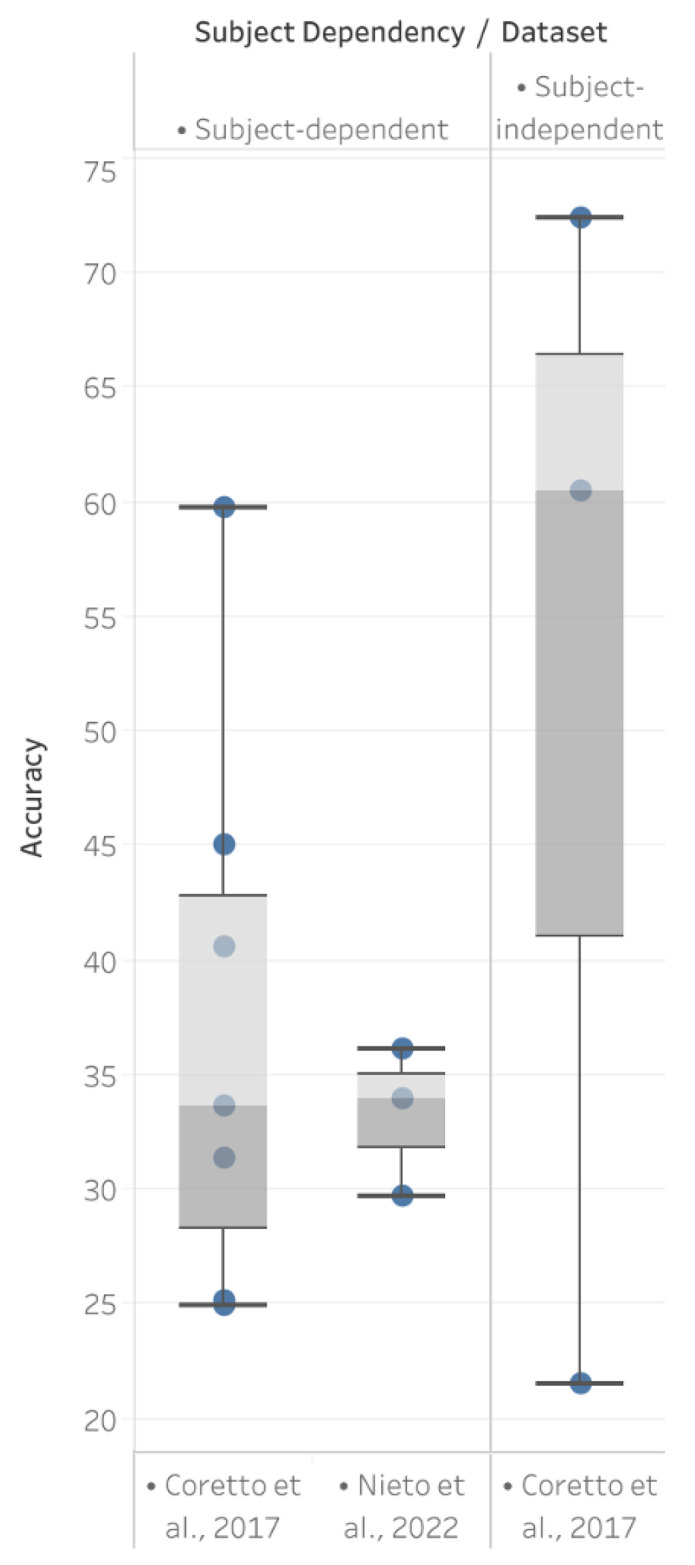
Boxplot illustrating accuracies based on subject dependency across studies using the datasets Nieto et al. (2022) [29] and Coretto et al. (2017) [28]. The data are derived from the following studies: Berg et al. (2021) [25], Biswas et al. (2018) [27], Biswas et al. (2022) [21], Cooney et al. (2020) [24], Dash et al. (2022) [16], García-Salinas et al. (2018) [22], Gasparini et al. (2022) [26], Kamble et al. (2022) [19], Kamble et al. (2023) [20], Lee et al. (2020) [17], Lee et al. (2021) [18] and Merola et al. (2023) [23].

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
