# Peer review of "Systematic Review of EEG-Based Imagined Speech Classification Methods"

_sensors, 2024, doi:10.3390/s24248168_

Round 1

Reviewer 1 Report

Comments and Suggestions for Authors

11.      In the introduction section, has any similar review paper been published? It will be useful for the reader to know what are the published related review papers and what the limitations are, meaning what key points were not addressed or comprehensively discussed in those published review papers. This is to let the readers know your motivation to write this review paper.

2.      Appendix Table A1 should be in the main text instead of in the appendix.

-        The second row has no reference number.

-        How do you rank the order of Table A1? Will that be easier to rank the papers by the alphabet of the authors, to be easier to match with Figure 2?

-        Some work only performed a 4-class classification. Which 2 directional words do they neglect?

3.      Line 350: Isn’t an accuracy of 33.9% is very low? So, this is not a binary classification?

4.      Figure 2: Is there any reason only a few studies have accuracy over 50%? Is there any reason that the accuracy of using the “Nieto et al. 2021” dataset is more uniform, but not the “Coretto et al. 2017” dataset?

5.      Line 595: “While subject-dependent models tend to achieve higher accuracy” does not match what is shown in Figure 3.

6.      Figure 3: The authors can give a more detailed explanation of the labels  (bars and dots) in the figure.

7.      More comparisons between the papers listed in this review article can be added. Now, most paragraphs individually describe a single paper only. A more integrated discussion to compare the technologies applied in each paper can be more useful for the readers.

Author Response

Comment 1: In the introduction section, has any similar review paper been published? It will be useful for the reader to know what are the published related review papers and what the limitations are, meaning what key points were not addressed or comprehensively discussed in those published review papers. This is to let the readers know your motivation to write this review paper.

Response1: Thank you for pointing this out. To address this, we have revised the introduction section to explicitly discuss related review papers, such as A state-of-the-art review of EEG-based imagined speech decoding, Decoding covert speech from EEG—a comprehensive review, and Advances in brain-computer interface for decoding speech imagery from EEG signals: a systematic review, highlighting their key contributions and limitations. We have clarified that this review paper uniquely focuses on publicly available EEG datasets featuring directional words, recent advancements between 2018 and 2023, and challenges in subject-independent approaches, multiclass scalability, and dataset limitations. These updates can be found in the revised manuscript on pages [2 - 3], introduction section, paragraph [5]

Comment 2: Appendix Table A1 should be in the main text instead of in the appendix.

  • The second row has no reference number.
  • How do you rank the order of Table A1? Will that be easier to rank the papers by the alphabet of the authors, to be easier to match with Figure 2?
  • Some work only performed a 4-class classification. Which 2 directional words do they neglect?

Response 2: Thank you for your suggestions.

  • Table A1 Placement: The table has been moved to the main text, specifically at the end of the Results section, as per your suggestion pages [16 - 17].
  • Missing Reference Number: The missing reference number has been corrected.
  • Ranking: The table was ordered by date; now the table is being rearranged alphabetically by the authors' names to align with Figure 2 for easier cross-referencing.
  • 4-Class Classification: it depends on the dataset, in Nieto dataset. there's only 4 words, while in Coretto data set there're 6

Comment 3: Line 350: Isn’t an accuracy of 33.9% very low? So, this is not a binary classification?
Response 3: Thank you for raising this point. We have clarified in the manuscript that the reported accuracy of 33.9% is for a multiclass classification task involving four classes, not a binary classification. This distinction has been explicitly stated in the relevant section on page [9], line [382], to avoid confusion.

Comment 4: Figure 2: Is there any reason only a few studies have accuracy over 50%? Is there any reason that the accuracy of using the “Nieto et al. 2021” dataset is more uniform, but not the “Coretto et al. 2017” dataset?

Response 4:The low accuracy (with few studies over 50%) is a result of the complexity of imagined speech tasks and inherent variability in EEG signals. Imagined speech involves subtle cognitive processes without physical expression, making a meaningful neural signal weak and challenging to detect. Also, EEG signals further complicate classification due to their low signal-to-noise ratio and susceptibility to artifacts such as muscle activity and environmental noise. These issues make it hard to isolate the specific neural signal related to imagined speech.

For the datasets, Nieto et al., (2021) is a newer dataset with fewer experiments on it (only 3 up to the date of this review). Therefore, there is less diversity in the preprocessing, feature extraction techniques, and classification models applied, leading to more consistent (lower) accuracy, while Coretto et al., (2017) have been tested more extensively, with 9 studies included in this review.

Comment 5: Line 595: “While subject-dependent models tend to achieve higher accuracy” does not match what is shown in Figure 3.
Response 5: Thank you for catching this inconsistency. The statement has been corrected to align with the findings shown in Figure 3. It now states that subject-dependent models achieve narrower accuracy ranges but do not consistently outperform subject-independent models in all cases. The revised text can be found on sub-section 4.2, page [18], line [679].

Comment 6: Figure 3: The authors can give a more detailed explanation of the labels  (bars and dots) in the figure.

Response 6:Thank you for the suggestion. We have revised the figure legend and the accompanying text to provide a clearer and more detailed explanation of the labels in Figure 3. Specifically, the updated description now clarifies the meaning of each bar, whisker, and dot, highlighting their representation of subject dependency, dataset variability, and accuracy distribution. These enhancements are reflected on page [14] lines [617 - 629]

Comment 7: More comparisons between the papers listed in this review article can be added. Now, most paragraphs individually describe a single paper only. A more integrated discussion to compare the technologies applied in each paper can be more useful for the readers.
Response 7:
We appreciate this feedback. We have revised the Results section to include integrated comparisons between studies at the end of each subsection. at the end of {Effect of Feature Extraction on Classification Performance}, {Classification Methods to Enhance Classification Performance}, {Investigation of Brain Lateralization in Imagined Speech},{Generalization and Scalability in EEG-Based Imagined Speech}

For instance, we now compare the performance of traditional machine learning methods (e.g., SVM) versus deep learning techniques (e.g., CNN and BiLSTM), highlighting their relative strengths and weaknesses in handling EEG data. Additionally, the impact of different preprocessing and feature extraction methods has been discussed in a comparative manner to provide a more holistic understanding of the field. These revisions are on pages [8 -12]

Reviewer 2 Report

Comments and Suggestions for Authors

The article aims to investigate existing scientific publications on the topic of “EEG-based Imagined Speech Classification Methods”. 

The authors outline the aim of the article as follows: “to establish a more methodologically robust and universal research path for developing a BCI system as a universal technology”.

The study closes the gap as to what methods are currently available for classification and what methods need to be developed in the near future.

The conclusions are consistent with the evidence and arguments presented.

The references are appropriate. 

I have a few recommendations for the authors of the article:

1. It can be seen that the authors used generative text in preparing the article. All GPT-like fragments should be reformatted by the authors.  For example, “Author (s) information: Identify the researchers who are the authors ...”. There are a lot of such fragments, and this should be corrected by all means.

2 The authors should make the purpose of the study more specific.

Author Response

Comment 1:
It can be seen that the authors used generative text in preparing the article. All GPT-like fragments should be reformatted by the authors. For example, “Author (s) information: Identify the researchers who are the authors ...”. There are a lot of such fragments, and this should be corrected by all means.

Response 1:
Thank you for pointing this out. We agree with this comment. Therefore, we have thoroughly reviewed the manuscript and reformatted all instances of generative text to align with a formal academic writing style. Specifically, phrases like what you mentioned have been revised for coherence and clarity. 

The phrase has been replaced with “The authors of each study were identified to recognize their contributions and understand the diversity of the research community." Other similar fragments across the manuscript have been updated to ensure consistency and alignment with professional writing standards.

Changes can be found in pages [5 - 6], sub-sections [2.4 - 2.5]

Comment 2:
The authors should make the purpose of the study more specific.

Response 2:
Totally Agree. We have accordingly revised the manuscript to make the purpose of the study more specific. the introduction section has been revised on pages [2 - 3], paragraphs [4 - 5- 6 -7 -8- 9]

Additionally, other relevant review papers were incorporated into the discussion to provide a broader context, and the gaps were identified 

"The purpose of this study is to address these gaps by systematically reviewing the latest advancements in EEG-based imagined speech classification between 2018 and 2023. It focuses on studies utilizing publicly available EEG datasets, particularly those featuring directional words such as 'up,' 'down,' 'left,' 'right,' 'forward,' and 'backward.' These directional words are pivotal for developing intuitive and efficient BCIs that facilitate navigation and communication for individuals with severe motor disabilities. Unlike previous reviews, this paper uniquely emphasizes the methodological and practical challenges associated with these datasets, providing a focused analysis of state-of-the-art techniques and limitations."

Round 2

Reviewer 1 Report

Comments and Suggestions for Authors

I have no more comments and suggestions.

Author Response

Thank you for taking the time to review our manuscript. We greatly appreciate your feedback and are pleased that no further comments or suggestions are required. Your insights have been invaluable in refining our work.

Reviewer 2 Report

Comments and Suggestions for Authors

Dear authors, thank you for considering all my comments.

Author Response

Thank you for your thoughtful comments and constructive feedback. We appreciate the time and effort you dedicated to reviewing our manuscript, and your insights have greatly improved the quality of our work.